# REINDIR: Repeated Embedding Infusion for Neural Deformable Image Registration

**Louis D. van Harten**[1,2]                          L.D.VANHARTEN@AMSTERDAMUMC.NL

**Rudolf L. M. van Herten**[1,2]                      R.L.M.VANHERTEN@AMSTERDAMUMC.NL

**Ivana Išgum**[1,2,3]                                 I.ISGUM@AMSTERDAMUMC.NL

[1] *Department of Biomedical Engineering and Physics, Amsterdam University Medical Center - location University of Amsterdam, the Netherlands.*

[2] *Informatics Institute, University of Amsterdam, the Netherlands.*

[3] *Department of Radiology and Nuclear Medicine, Amsterdam University Medical Center - location University of Amsterdam, The Netherlands.*

**Editors:** Accepted for publication at MIDL 2024

## Abstract

The use of implicit neural representations (INRs) has been explored for medical image registration in a number of recent works. Using these representations has several advantages over both classic optimization-based methods and deep learning-based methods, but it is hindered by long optimization times during inference. To address this issue, we propose REINDIR: Repeated Embedding Infusion for Neural Deformable Image Registration. REINDIR is a meta-learning framework that uses a combination of an image encoder and template representations, which are infused with image embeddings to specialize them for a pair of test images. This specialization results in a better initialization for the subsequent optimization process. By broadcasting the encodings to fill our modulation weight matrices, we greatly reduce the required size of the encoder compared to approaches that predict the complete weight matrices directly. Additionally, our method retains the flexibility to infuse arbitrarily large encodings. The presented approach greatly improves the efficiency of deformable registration with INRs when applied to in-distribution data, while remaining robust to severe domain shifts from the distribution the method is trained on.

**Keywords:** Deformable image registration, meta-learning, implicit neural representations.

## 1. Introduction

Deformable image registration seeks to transform pairs of images to a shared coordinate space by identifying a deformation vector field (DVF) that maximizes the local correspondences of image content. Many medical image processing pipelines rely on robust image registration as a precursor to any downstream analyses. With the rise of deep learning, convolutional neural network (CNN) approaches have been proposed for this paradigm (Fu et al., 2020). The inductive bias inherent to CNNs results in a grid-based DVF approximator: once trained, such methods directly predict a rasterized DVF given a set of new, unseen images (de Vos et al., 2019). Although providing fast approximations, CNN-based methods are highly sensitive to domain shifts resulting from out-of-distribution data. This may be mitigated by optimizing an image registration similarity metric at test-time (Fechter and Baltas, 2020), though this is typically slow and undesirable for large models used in learning-based methods.

In recent years, *implicit neural representations* (INR) have been proposed for deformable image registration (Wolterink et al., 2022). Within this paradigm, a neural network is leveraged to approximate the DVF by mapping input coordinates to their respective deformation vectors. This optimization framework has several advantages over both classic optimization-based methods and deep learning-based methods, such as precise regularization of spatial derivatives, robustness to domain shift, and the ability to capture small details in the DVF, but it is hindered by long optimization times during inference. This makes its use impractical in clinical workflows, especially in cases where either speed is of the essence, such as for stroke patients or image-guided therapy, or in cases where registration over many image acquisition time points is required, such as intestinal or cardiac motion analysis.

Several methods have been proposed to address the issue of test-time optimization of INRs. One such method is meta-learning, which focuses on learning an improved INR parameter initialization from a training set. At test time, the INR is then able to quickly adapt to new similar tasks (Nichol et al., 2018; Tancik et al., 2021). This principle has recently been applied for medical image registration (Baum et al., 2023). Another effective strategy for optimizing INRs is to infuse the model parameters with a task-specific prior. This approach is exemplified in models like DeepSDF, where an auto-decoder architecture is used to optimize a latent vector at test time while freezing the remaining INR weights (Park et al., 2019). An alternative approach is the use of hyper-networks, in which an additional neural network is employed to directly predict the weights or modulate the activations of an INR. This immediately provides the INR with a useful bias, allowing it to rapidly adapt to the task at hand (Ha et al., 2017; Perez et al., 2018; Babu et al., 2023).

Given these recent developments in INR optimization, we introduce an accelerated method for the task of deformable image registration. Specifically, we propose REINDIR: Repeated Embedding Infusion for Neural Deformable Image Registration. In addition to improving the optimization speed via meta-learning, we condition the initialization of INR parameters on a prior in the form of a repeated encoding of the to-be-registered image pair. We demonstrate the effectiveness of our method in the 3D lung CT registration paradigm, as well as on the task of cone-beam CT-guided radiation therapy.

## 2. Methods

To accelerate the optimization process of INRs for deformable registration, we propose REINDIR: Repeated Embedding Infusion for Neural Deformable Image Registration. The method generates INRs that are infused with a learned embedding of the input images. We infuse these embeddings into the neural representations by broadcasting them into a modulation matrix, which has the same shape as the weight matrices of each hidden layer. The modulation matrices for each layer are applied to the weight matrices of template INRs, which are trained in conjunction with the encoder that produces the embeddings. These template INRs can be thought of as implicit decoders for the embeddings. An overview of the proposed method is presented in Figure 1.

### 2.1. Implicit neural representations

In the paradigm of implicit neural representations, neural networks are specifically leveraged for their capability as universal function approximators. Rather than operating on image

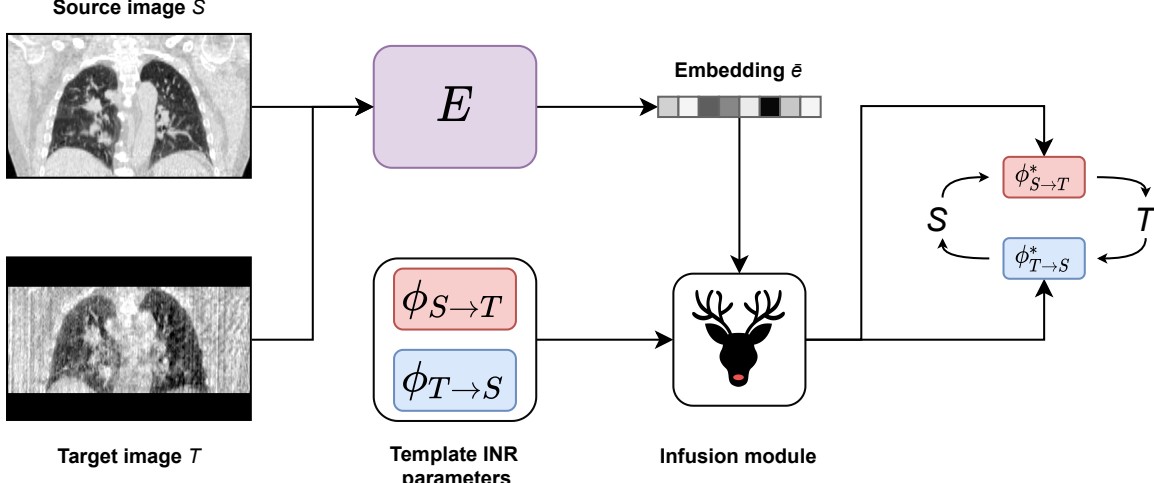

Figure 1: Overview of the proposed method. A source image $S$ and target image $T$ are embedded into a combined representation $\bar{e}$ using a trainable convolutional encoder $E$. We subsequently infuse the parameters of two template implicit neural representations (INR) $\phi_{S \to T/T \to S}$ by modifying their weights using the embedding $\bar{e}$ in the infusion module. This results in two task-informed INRs ($\phi^*$) capable of rapidly fitting the deformable registration of coordinates in $S$ to coordinates in $T$ in a cycle-consistent fashion.

values, the INR processes continuous coordinates which are subsequently mapped to some output value. For image analysis tasks, a loss function is typically included which relates outputs to image information. The image information therefore only enters the network through gradient backpropagation.

For the task of image registration, a neural representation in the form of $\Phi(\bar{x}) = \upsilon(\bar{x}) + \bar{x}$ is employed (Wolterink et al., 2022). Here, $\upsilon(\bar{x})$ denotes a sinusoidal representation network (SIREN) mapping input coordinates $\bar{x}$ from the source domain $S$ to deformation vectors (Sitzmann et al., 2020). By applying the vectors $\upsilon(\bar{x})$ to $\bar{x}$, the corresponding coordinates in the target domain $T$ can be inferred. Specifically, we employ cycle-consistent INRs (ccIDIR) as presented in (van Harten et al., 2024) for more accurate and more robust registration results. As such, a set of INRs is trained to perform both the mapping $S \to T$ and $T \to S$ by minimizing an additional cycle-consistency term in the loss function.

## 2.2. Meta-learning and embedding infusion

Since random initialization of INRs is central to their slow test-time optimization, meta-learning strategies such as Reptile (Nichol et al., 2018) have been proposed. In this paradigm, the initial parameters of an INR are optimized by fitting the network to batches of examples in a training dataset. The initial state of the network is subsequently updated by taking a small step into the direction of the fully optimized parameters. As such, the

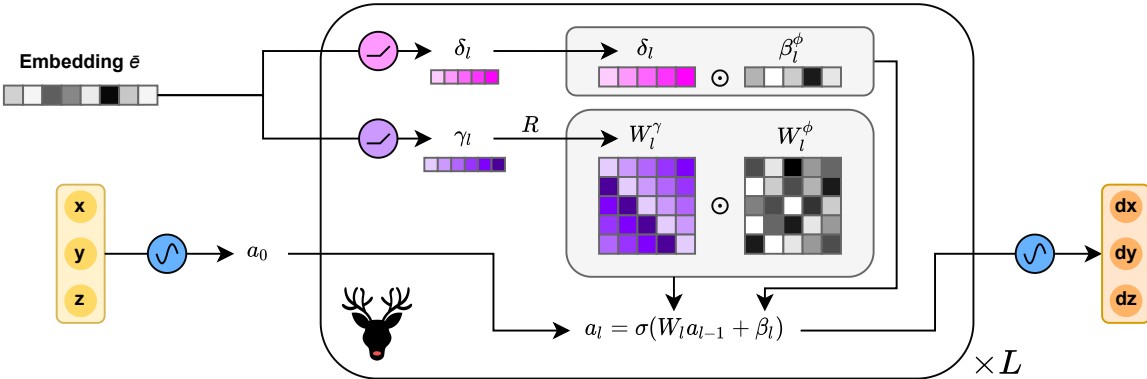

Figure 2: The REINDIR infusion module. For each implicit neural representation (INR) layer $l$, two modulation vectors $\gamma_l$ and $\delta_l$ are inferred from $\overline{e}$. In order to perform pointwise modulation of weight matrix $W_l^\phi$, the $\gamma_l$ vector is broadcasted to $W_l^\gamma$ through repetition ($R$). Meanwhile, bias $\beta_l^\phi$ is directly modulated by the same-size vector $\delta_l$. This allows the model parameters $W_l$ and $\beta_l$ to instantly incorporate motion pattern information between source and target images. This is repeated for all $L$ layers of each INR. Note that no modulation matrices or bias vectors are predicted for the first $(x, y, z \to a_0)$ and final $(a_L \to dx, dy, dz)$ INR layers.

INR learns a generalizable template for function approximation that can rapidly fine-tune to new instances from the same distribution. However, as the template is shared between all examples in the training set, the benefit of this approach is bounded by the dissimilarity of different examples within the target distribution.

To address this, we introduce an image encoder that creates an embedding from the concatenation of the source and target images. By infusing the template INRs with this embedding, we can leverage the image encoder as a hyper-network for specializing the initializations to each image pair. As illustrated in Figure 2, we achieve this by predicting two modulation vectors ($\gamma_l$ and $\delta_l$) for each INR layer, which modulate the weight matrix $W_l^\phi$ and the bias vector $\beta_l^\phi$ respectively. Modulation of $W_l^\phi$ is achieved by repeating and subsequently reshaping $\gamma_l$ to fit $W_l^\phi$, allowing for pointwise multiplication. Meanwhile, $\beta_l^\phi$ is directly modulated by the same-size vector $\delta_l$. The INRs used in this work are SIRENs (Sitzmann et al., 2020) with three hidden layers containing 256 features each. Modulation is performed for the two weight matrices between these hidden layers.

During training, an image pair is first embedded by a convolutional encoder based on the LKU-Net architecture (Jia et al., 2022), which is used to generate modulation vectors. INRs are subsequently optimized for a number of iterations using this embedding, after which the image encoder is updated. This allows us to train both the template INRs and the image encoder end-to-end. The full training procedure is detailed in Algorithm 1. Additional implementation details are included in Appendix A.

---

**Algorithm 1:** REINDIR training loop

---

**Data:** $D_{DIRLAB}$, $D_{NLST}$, $D_{CBCT}$

**Input:** $E$, $\phi_{S \to T}^o$, $\phi_{T \to S}^o$

**for** $iteration = 1, 2, 3, \ldots, N_{outer}$ **do**

    $\delta_{INRs} = 0$                         $\triangleright$ $\delta$ between parameters of $\phi^o$ and $\phi$

    **for** $set$ in $[D_{DIRLAB}, D_{NLST}, D_{CBCT}]$ **do**

        $\phi_{S \to T}, \phi_{T \to S} \leftarrow \phi_{S \to T}^o, \phi_{T \to S}^o$

        $S, T \leftarrow \textsc{Sample}(set)$

        $\bar{e} \leftarrow \textsc{RunFrozenEncoder}(S, T, E)$

        **for** $iteration = 1, 2, 3, \ldots, N_{inner}$ **do**

            $\phi_{S \to T}, \phi_{T \to S} \leftarrow \textsc{UpdateINRs}(S, T, \bar{e}, \phi_{S \to T}, \phi_{T \to S})$

        **end**

        $E \leftarrow \textsc{UpdateEncoder}(S, T, E, \phi_{S \to T}, \phi_{T \to S})$

        $\delta_{INRs} \leftarrow \delta_{INRs} + \frac{(\phi_{S \to T}^o, \phi_{T \to S}^o) - (\phi_{S \to T}, \phi_{T \to S})}{N_{sets}}$

    **end**

    $\phi_{S \to T}^o, \phi_{T \to S}^o \leftarrow (\phi_{S \to T}^o, \phi_{T \to S}^o) + \delta_{INRs}$

**end**

---

## 3. Data

In this work, a total of three datasets are used for two different registration tasks. All scans from all sets were resampled to an isotropic 1 $mm^3$ using trilinear interpolation.

The first two datasets focus on the task of 4D lung CT inspiration-to-expiration image registration. We include data from the DIR-LAB (10 patients) and NLST (110 patients) studies (Castillo et al., 2009; Aberle et al., 2011). The DIR-LAB dataset contains 300 manually annotated lung landmarks per scan, which serve as a ground truth for determining the registration error. For the NLST data, we use the subset made available for the Learn2Reg challenge (Hering et al., 2022). For 100 out of 110 cases in this set, a varying number of automatically generated matched keypoints are made available.

In the third dataset, we focus on registration between pre-therapeutic fan-beam CT (FBCT) and interventional low-dose cone-beam CT (CBCT). The dataset made available for this task is also part of the Learn2Reg challenge, and features a total of 14 patients (Hugo et al., 2016). For each patient, a planning FBCT as well as two CBCT scans are made available, of which the first CBCT scan was acquired at the beginning of therapy and the second one at the end of therapy. The challenge considers the registration of all visible thoracic organs, and is required to be both fast and accurate to minimize the radiation dose for organs at risk. Automatically generated keypoints for the first 11 patients were provided by the challenge organizers. However, upon visual inspection, correspondence between keypoints in this set proved to be limited. We therefore generate corresponding keypoints in this set by optimizing image pairs using an existing registration method (van Harten et al., 2024) within automatically generated lung masks (Hofmanninger et al., 2020)[1]. Two image pairs were excluded in this process due to high variability in matched keypoints after optimization. Additional details are included in Appendix B.

---

1. Generated keypoints are available at https://github.com/louisvh/cbct_generated_keypoints

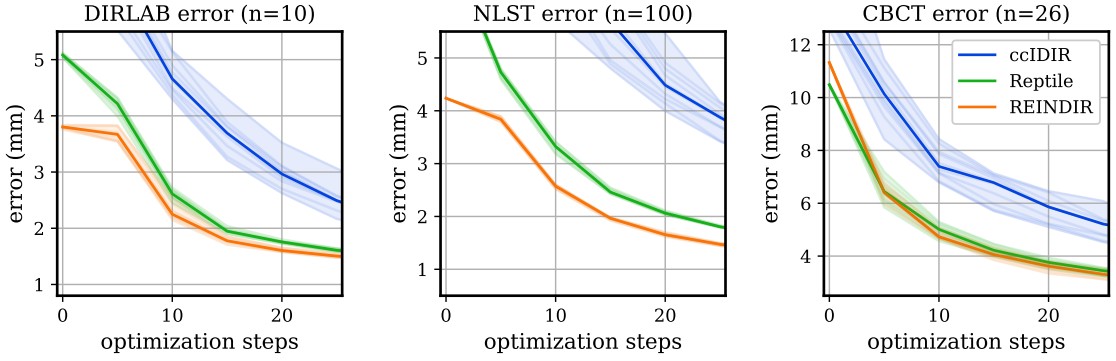

Figure 3: Registration error over time, comparing random initialization (ccIDIR), learned initialization (Reptile) and the proposed learned initialization with embedding infusion (REINDIR). Median results plotted for 10 different random seeds; solid lines indicate the median, shaded areas indicate the range among seeds. Extended results with performance and runtime benchmarks are included in Appendix C.

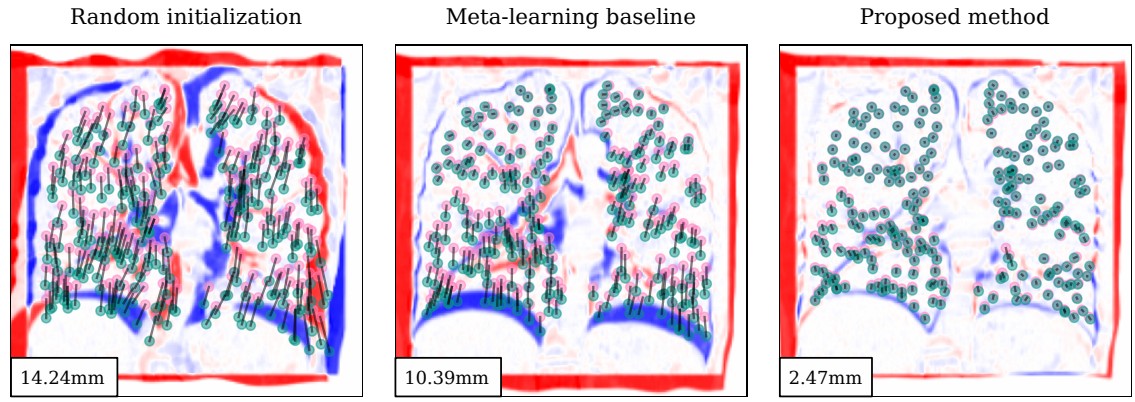

Figure 4: Qualitative comparison of zero-shot registration error at initialization in an image from the NLST set before optimization. A subset of the matched landmarks for the inspiration and expiration phases is shown in teal and pink respectively, connected by black lines. Bottom left corners show the mean landmark error.

## 4. Experiments and Results

We compare our proposed initialization to ccIDIR with random initialization as proposed in (van Harten et al., 2024), as well as to a Reptile initialization (Nichol et al., 2018), which is equivalent to using trained template INRs without embedding infusion. Since both Reptile and REINDIR are learning-based methods that fit a training set distribution, we present

Figure 5: Example of a failure case from the CBCT set, showing error without registration, error before optimization and after 25 optimization steps. The encoder failed to generalize to this image pair, resulting in a low-quality initial registration. A subset of the matched landmarks for the inspiration and expiration phases is shown in teal and pink respectively, connected by black lines. Bottom left corners show the mean landmark error.

our results for 5-fold cross-validation. Specifically, for each fold the methods are optimized using images from all sets. Methods are compared in terms of the landmark registration error, i.e. the mean error for keypoints between image pairs.

A comparison of quantitative results for INR-based deformable image registration methods is presented in Figure 3. For each image pair, the method is optimized within a near-instant timeframe (25 iterations $\sim$ 1 s, Nvidia RTX 2080Ti). For both the DIRLAB and NLST sets, the embedding prior provided by REINDIR produces a lower registration error at initialization, resulting in faster convergence. For the CBCT set, Reptile and REINDIR perform similarly, with both methods converging faster than the ccIDIR baseline. Errors in this set were typically higher due to the more complex registration task of noisy CBCT data. It should further be noted that registration errors for all methods eventually converge to a similar value given a large number of test-time optimization steps.

An example image from the NLST set is shown in Figure 4, which compares the proposed initialization with random initialization and Reptile initialization, which is equivalent to using trained template INRs without embedding infusion. In this example, the proposed method is able to instantly correct the general motion pattern without any INR update steps. We further exemplify a case of the CBCT set for which REINDIR initialization was not beneficial in Figure 5. Additional examples are included in Appendix D.

## 5. Discussion and Conclusion

We have presented a method for deformable image registration with implicit neural representations that explicitly incorporates image information from to-be-registered image pairs.

We embed image information into the INR by modulating the weights and biases of the network, which can prove beneficial for the initialization of the registration task.

As illustrated in Figure 3, our proposed method provides a useful bias in the registration of both the DIRLAB and the NLST set, for which the keypoint registration error is significantly lower upon initialization of the INR. This trend is however not visible in the results of the CBCT set, for which the optimization process of REINDIR closely follows Reptile. We hypothesize that the low number of available image pairs featured in this set may hamper the ability of the convolutional encoder to generalize to unseen data. This is further complicated by the fact that the CBCT images only provide a limited view of the thorax while being highly variable in terms of appearance and noise level. Despite this, REINDIR never negatively impact registration performance, as illustrated in Figure 5. Though the initialization fails to capture the registration motion pattern, the network is able to rectify the deformation given a small number of optimization steps, resulting in a performance similar to Reptile.

The results further indicate that REINDIR performance is correlated with training set size: the best results were observed for the NLST set, for which in the cross-validation setting 88 image pairs were available during training. In comparison, only 8 and 20 training image pairs were available for the DIRLAB and CBCT sets, respectively. In line with expectations, performance gain for DIRLAB is therefore less pronounced, and is limited in the CBCT set. Future work may therefore include a large set of more diverse training data, or research more expressive encoding architectures. This could further boost zero-shot performance and limit the necessity of test-time optimization to out-of-distribution data.

An important distinction with recent work on INR-based methods is that our method does not use hash encoding (Müller et al., 2022). Hash grids are used to encode the input space, taking advantage of the sparse nature of data in many natural image processing tasks. The aliasing in these hash grids compresses the problem such that subsequent MLPs can be smaller. However, hash grids introduce discontinuities in the spatial encoding (Huang and Alkhalifah, 2024), which complicates the optimization of image registration INRs, as this requires computation of spatial derivatives throughout the valid region of the input space.

It should further be noted that modulation of the weight matrix is not the only option for INR manipulation. Recent works (Dupont et al., 2022; Papa et al., 2023) proposed to modulate INR activations rather than the weight matrices themselves, which allows the network to condense meaningful sample-specific information into a low number of parameters. Future work may investigate such modulation variations. A possible disadvantage of such an approach is that it is not possible to modulate the interactions between different features. Additionally, the amount of information that can be infused into the template network is limited by the number of features in each layer, which is a very small fraction of the number of weights in those layers. In REINDIR, we keep the length of the encodings a free parameter, which allows the amount of information infused into the template representations to be chosen without such constraints. Future work may further explore the impact of different encoding lengths and types of embedding infusion.

In conclusion, this study presented a method for accelerated medical image registration by infusing implicit neural representations with embeddings from to-be-registered image pairs. Our experiments show how this prior can help guide the registration process and produce useful initialization for deformable image registration.

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

## Appendix A. Implementation details

### A.1. Encoder architecture

The image encoder used in this work is based on the encoder of the LKU-Net$_{6,5}$ architecture (Jia et al., 2022). The authors of the paper extensively explore differences between convolutional and transformer-based backbones, and are listed as one of the best-performing algorithms on the Learn2Reg NLST leaderboard. Nevertheless, we expect any sufficiently parameterized model to perform well as an image encoder. LKU-Net employs a convolutional encoder where some of the convolution layers consist of three parallel residual paths with convolution kernels of size 1, 3 and 5. The results from these convolutions are summed together with the input and passed to a PReLU activation (He et al., 2015). At the encoder bottleneck, we append a set of three convolutional layers to compress the representation into embedding $\bar{e}$. The penultimate layer of the encoder uses non-isotropic kernels to compensate for our use of non-isotropic patches (i.e. $[336, 288, 336]$). In this work, we used $\bar{e} \in \mathbb{R}^{1025}$, but the length of this embedding can be changed to control the amount of information that can be infused into the INRs, as appropriate for the target task. We leave an analysis on the impact of the length of this embedding for future work.

The modulated INRs used in this work are SIRENs (Sitzmann et al., 2020) with three hidden layers, each containing 256 features. The two weight matrices between these hidden layers are modulated in REINDIR; we do not modulate the input or the output layers. To generate the modulation weights and biases, we add two linear heads to the encoder for each modulated layer. The encoder architecture is shown in Table A.1.

Table 1: Convolutional Encoder Architecture

| Layer | Features | Kernel Size | Stride |
|:---:|:---:|:---:|:---:|
| 1 | 6 | 3 | 1 |
| 2 | 6 | 3 | 1 |
| 3 | 12 | 3 | 2 |
| 4 | $12 \times 3$ | $1, 3, 5$ | 1 |
| 5 | 24 | 3 | 2 |
| 6 | $24 \times 3$ | $1, 3, 5$ | 1 |
| 7 | 48 | 3 | 2 |
| 8 | $48 \times 3$ | $1, 3, 5$ | 1 |
| 9 | 48 | 3 | 2 |
| 10 | $48 \times 3$ | $1, 3, 5$ | 1 |
| 11 | 96 | 3 | 3 |
| 12 | 96 | $[3, 2, 3]$ | 1 |
| 13 | 1025 | 5 | 1 |
| $\delta_l \mid l \in L$ | 256 | 1 | 1 |
| $\gamma_l \mid l \in L$ | 1025 | 1 | 1 |

## A.2. Training and test-time optimization

The method is trained using the procedure described in Algorithm 1. The training parameters are listed in Table A.2. The optimization procedure for the inner loop closely resembled the procedure described in (van Harten et al., 2024), with one notable exception: during training, SGD was used as the optimizer for the template INRs. Training with Adam as the optimizer for the inner training loop caused instability in the training procedure. As this does not impact the test-time optimization, Adam is used during inference. With the exception of the embedding length, parameters for the baseline methods were matched to the parameters used for the proposed method.

Table 2: Optimization parameters

| | | | | | |
|---|---|---|---|---|---|
| Patch size | | | $[336, 288, 366]$ | | |
| Embedding length $\bar{e}$ | | | 1025 | | |
| Iterations $N_{outer}$ | | | 22750 | | |
| Iterations $N_{inner}$ | | | 10 | | |
| Similarity loss | | | Normalized cross-correlation | | |
| Regularization losses | | | Symmetric Jacobian determinant | $\alpha = 0.05$ | |
| | | | Cycle consistency | $\alpha = 0.001$ | |
| Batch size | Training | Encoder | 1 | | |
| | | INRs | 5000 | | |
| | Inference | INRs | 10000 | | |
| Optimizer | Training | Encoder | Adam | Cosine annealing | $lr \leq 10^{-5}$ |
| | | INRs | SGD | Static learning rate | $lr = 10^{-4}$ |
| | Inference | INRs | Adam | Static learning rate | $lr = 10^{-4}$ |
| Scale augmentations | | | $0.8 - 1.2$ | | |
| Cosine annealing | Cycles | | 4 | | |
| | Warm-up | | 250 | | |

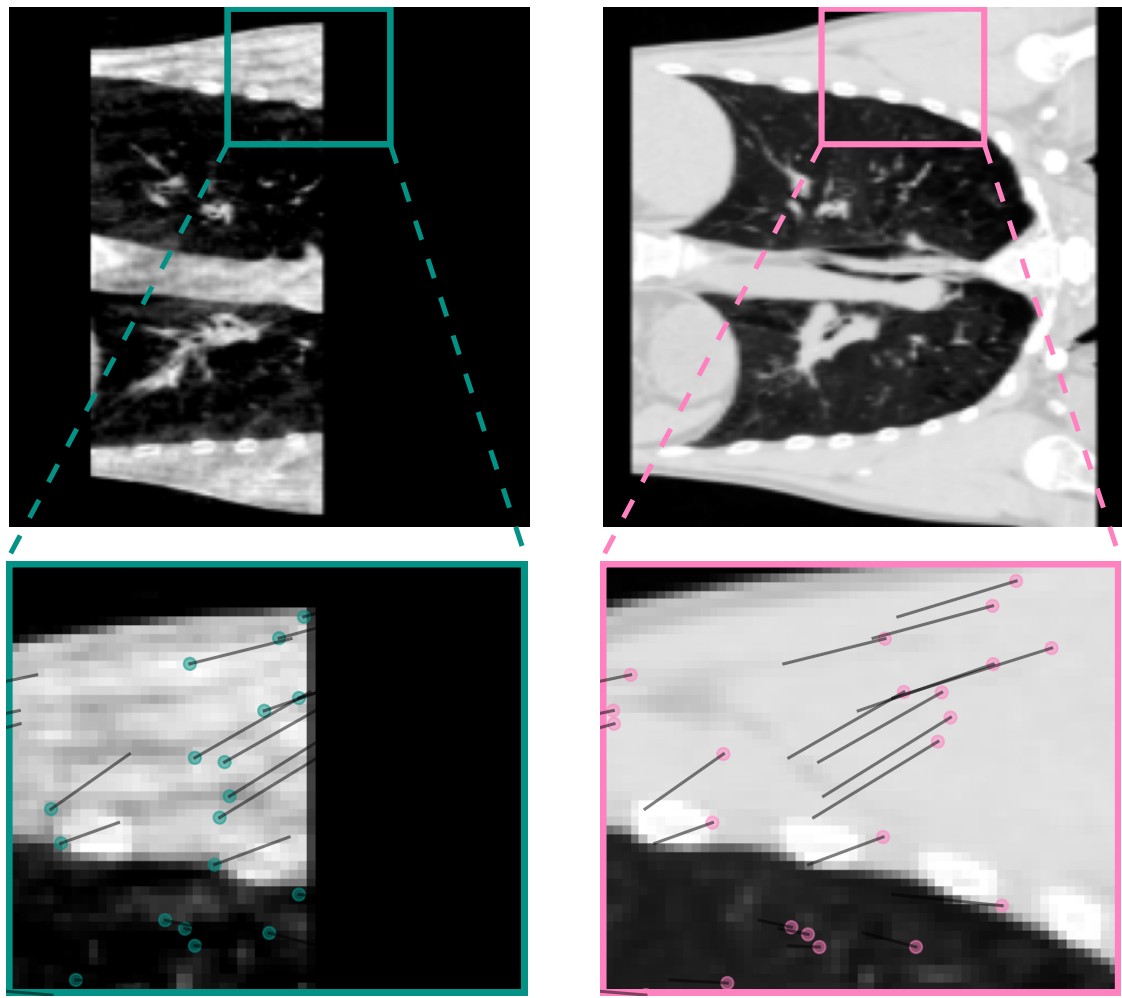

Figure 6: A visualization of the original ground-truth markers from the CBCT set. From visual inspection, many of these markers show large registration errors.

## Appendix B. CBCT marker definition

While the CBCT set from the Learn2Reg challenge provides matched keypoints that can be used for method validation, manual inspection of these markers revealed a large fraction of these markers contained large registration errors (as shown in Figure 6), presumably related to the artifacts present in the cone beam CTs. As a result, we considered these markers sub-optimal for method evaluation.

We extracted new reference markers in automatically generated lung masks of the cone beam CTs using a deep learning-based segmentation method (Hofmanninger et al., 2020) and we propagated these markers to the corresponding fan beam CTs using an ensemble of ten registration results obtained from a cycle-consistent implicit neural registration system

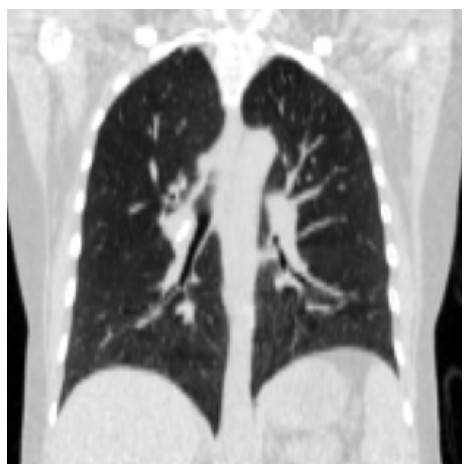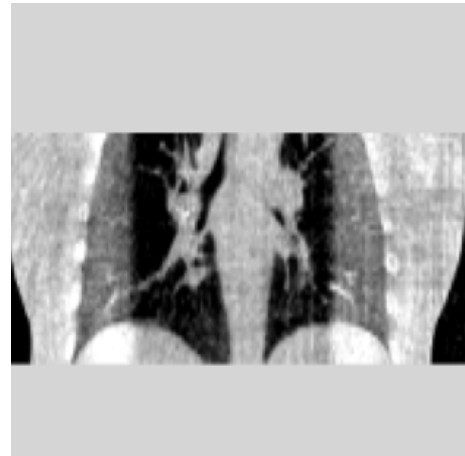

Figure 7: One of the image pairs that resulted in a mean uncertainty over 5mm in the fully converged registration instances. The image artifacts in the cone beam CT made this image pair unsuitable for registration with a global NCC loss.

optimized for 2500 iterations, as detailed in (van Harten et al., 2024). This registration method produces an uncertainty estimate resulting from the cycle-error after optimization. For each image pair, we ranked the registration results by average estimated uncertainty and propagated the markers using the five most confident instances. These instances were combined by calculating the median deformation vector for each marker.

For each propagated image pair, we evaluated the mean uncertainty of the five registration instances used for propagation. This revealed two outliers, which had mean uncertainties of more than 5mm. One of these image pairs is shown in 7. For these image pairs, the cone beam CTs contained artifacts that made them unsuitable for registration with the global NCC loss used in this work. Hence, we excluded these images from further analysis.

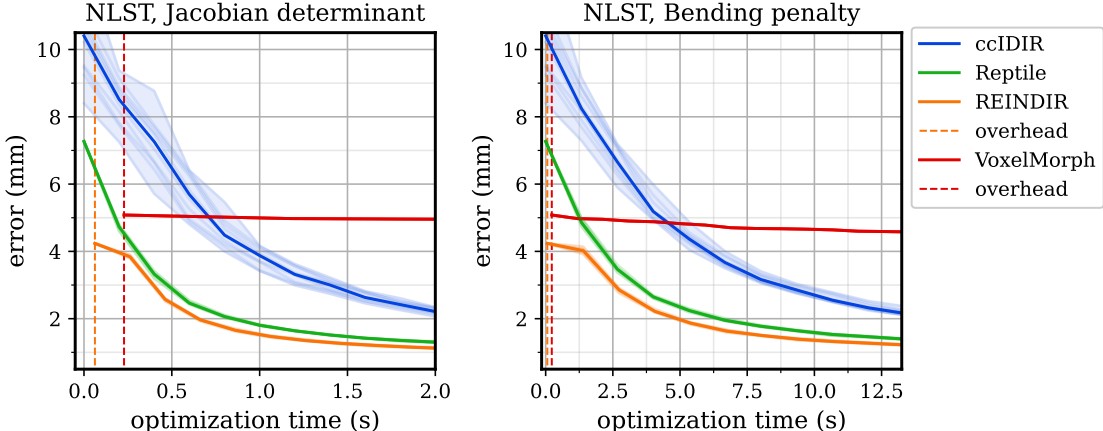

Figure 8: Registration error in the NLST set as a function of test-time optimization run-time. Results for the INR-based methods are reported for Jacobian determinant regularization (left) and Bending penalty regularization (right). The VoxelMorph baseline employs the L2 smoothness penalty provided by the original authors in both graphs. Dashed vertical lines indicate the additional overhead given the presence of a learning-based image encoder, which is present for both REINDIR (orange) and VoxelMorph (red). All experiments were performed on an Nvidia RTX 3090 GPU with an Intel Xeon Gold 6128 clocked at 3.7 GHz.

## Appendix C. Runtime analysis

Figure 8 provides an analysis of inference times for ccIDIR (van Harten et al., 2024), Reptile, REINDIR and a baseline VoxelMorph approach (Balakrishnan et al., 2019). The additional overhead of the encoder used in this work is minimal, it being equivalent to approximately 1.5 optimization steps when using Jacobian determinant regularization, or 0.23 steps when regularizing with a Bending penalty.

To include a learning-based image registration baseline, we provide the results on the NLST dataset for VoxelMorph (Balakrishnan et al., 2019) here. The method was trained using the the default settings of the author's PyTorch implementation on GitHub (commit: ca28315), employing seven diffeomorphic integration steps of the vector field. The only change was to reduce the network channel dimensions from 16 to 6, which was necessary to fit a set of full-size NLST images ($2 \times 336 \times 288 \times 366$) into GPU memory (3090, 24GB) during training and test-time optimization. This network was trained for 200 epochs in 5-fold cross-validation. At test-time, the to-be-registered image pair was repeatedly processed by the VoxelMorph pipeline, using the same training settings as used in the original training code; each optimization step took 0.95 seconds.

## Appendix D. Additional qualitative results

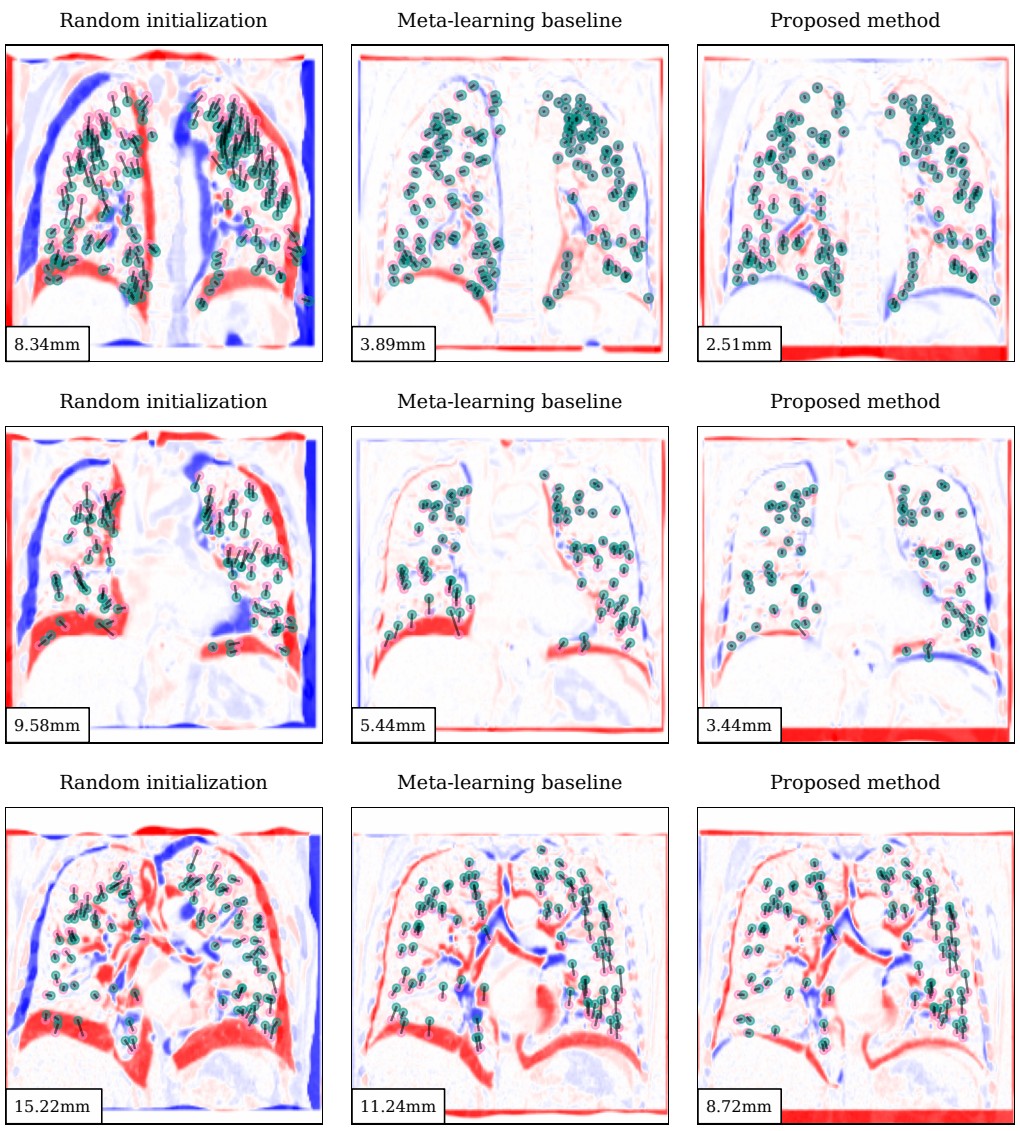

Figure 9: Qualitative comparison of zero-shot registration error for images from the DIRLAB set, showing the best result (top), the result closest to the median performance (middle) and the worst result (bottom) for the proposed method. A subset of the matched landmarks for the inspiration and expiration phases is shown in teal and pink respectively, connected by black lines. Bottom left corners show the mean landmark error.

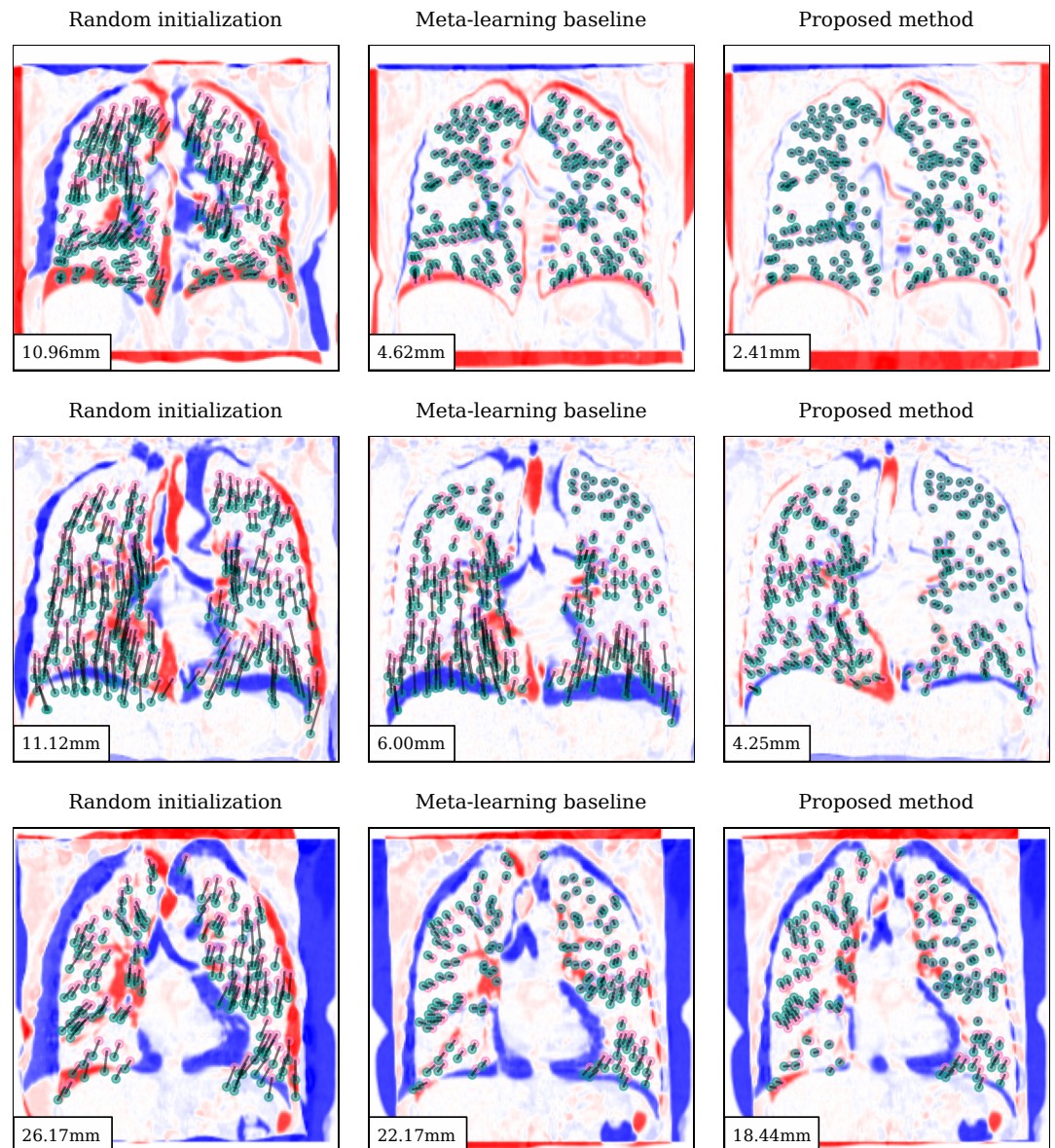

Figure 10: Qualitative comparison of zero-shot registration error for images from the NLST set, showing the best result (top), the result closest to the median performance (middle) and the worst result (bottom) for the proposed method. A subset of the matched landmarks for the inspiration and expiration phases is shown in teal and pink respectively, connected by black lines. Bottom left corners show the mean landmark error.

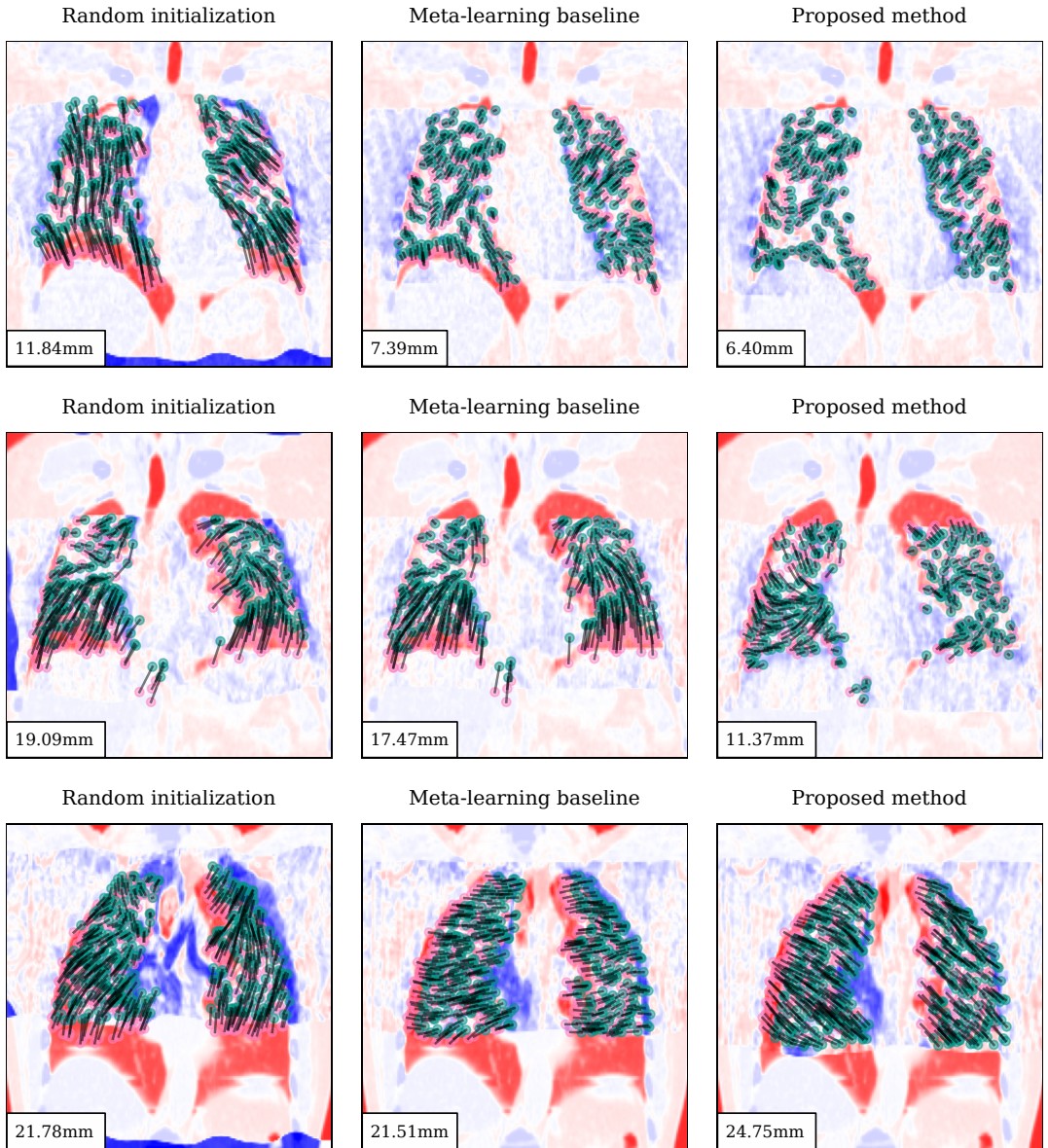

Figure 11: Qualitative comparison of zero-shot registration error for images from the CBCT set, showing the best result (top), the result closest to the median performance (middle) and the worst result (bottom) for the proposed method. A subset of the matched landmarks for the inspiration and expiration phases is shown in teal and pink respectively, connected by black lines. Bottom left corners show the mean landmark error.

