# OpenReview forum: "REINDIR: Repeated Embedding Infusion for Neural Deformable Image Registration"
_MIDL.io/2024/Conference — MIDL 2024 Poster_

### Official Review · Reviewer_KSKH · 2024-02-28

**Confidence:** 4
**Preliminary Rating:** 3
**Final Rating:** 2.5

**Summary:**

This paper proposes REINDIR: Repeated Embedding Infusion for Neural Deformable Image Registration. This meta-learning framework combines an image encoder and template representations that are infused with image embeddings to specialize them for image pairs.
The presented approach improves the efficiency of deformable registration with INRs and remains robust to domain shifts from the distribution the method is trained on.

**Strengths:**

- By broadcasting the encodings to fill modulation weight matrices, the proposed REINDIR reduces the required size of the encoder compared to approaches that predict the complete weight matrices, improving the efficiency of deformable registration with INRs.

- This meta-learning framework that combines an image encoder and template representations for registration is interesting.

**Weaknesses:**

- The submission lacks a comprehensive qualitative and quantitative comparison with current state-of-the-art (SOTA) registration methods, as well as a benchmark of runtime performance.
- The motivation behind using a meta-learning framework for the registration task is not well-defined, particularly in distinguishing it from fine-tuning approaches. The data presented in Figure 3 indicates comparable performance between Reptile and the proposed REINDIR, which calls for a more detailed justification.
- The methodology is not adequately elucidated, with Figures 1 and 2 failing to convey the nuances of the proposed approach.

**Detailed Comments:**

Please refer to the Weaknesses Part.

**Justification Of Final Rating:**

Some concerns have been addressed.
The additional evaluation comparison with VoxelMorph registration methods and updated figure captions help address my concerns about the experiment and the paper's clarity. However, the VoxelMorph is the SOTA method in 2019.
 While fine-tuning can be leveraged to optimize registration CNNs/Transformers to specific image pairs, the authors make clear that the encoder employed in their work only requires a single forward pass and is never tuned at test time. However, my initial concerns still stand, cause the encoder often contains meta-information rather than task-specific information which needs tunning.

**Justification Of The Preliminary Rating:**

Concerns about unconvincing experiment verification and unclear motivation.

- The submission lacks a comprehensive qualitative and quantitative comparison with current state-of-the-art (SOTA) registration methods, as well as a benchmark of runtime performance.
- The motivation behind using a meta-learning framework for the registration task is not well-defined, particularly in distinguishing it from fine-tuning approaches. The data presented in Figure 3 indicates comparable performance between Reptile and the proposed REINDIR, which calls for a more detailed justification.
- The methodology is not adequately elucidated, with Figures 1 and 2 failing to convey the nuances of the proposed approach.

**Questions To Address In The Rebuttal:**

Concerns about unconvincing experiment verification and unclear motivation.

---

> ### Author Response · Authors · 2024-03-16
> **Response to Reviewer KSKH (1/1)**
>
> We thank the reviewer for their comments, which we have addressed below.
>
> __W1:__ The submission lacks a comprehensive qualitative and quantitative comparison with current state-of-the-art (SOTA) registration methods, as well as a benchmark of runtime performance.
>
> __A:__ We agree that a comparison to a registration baseline could benefit the paper. Since there is unfortunately not enough space to include this in the main text, we have added an additional appendix in which we evaluate the performance of VoxelMorph (Balakrishnan et al., 2019) on our largest dataset, i.e. NLST. We present both a comparison with this method as well as runtime benchmarking performance in __Appendix C__. The figure shows VoxelMorph performance using a single forward pass (red dotted vertical line), as well as performance given fine-tuning steps in a test-time training setting (red line). The figure further shows the inference time for all three previously discussed INR-based approaches, and we present results for both Jacobian determinant regularization (left) and Bending energy penalty regularization (right). Finally, the orange dotted line represents the overhead in REINDIR resulting from the forward pass of Encoder _E_, which is equivalent to 1.5 steps for Jacobian determinant regularization and 0.23 steps for Bending penalty regularization.
>
> __W2:__ The motivation behind using a meta-learning framework for the registration task is not well-defined, particularly in distinguishing it from fine-tuning approaches. The data presented in Figure 3 indicates comparable performance between Reptile and the proposed REINDIR, which calls for a more detailed justification.
>
> __A:__ Fine-tuning can indeed be leveraged to optimize registration CNNs/Transformers to specific image pairs at test-time, as well. A downside to this approach is that it requires optimization of a large network, therefore eliminating the efficiency of such methods compared to optimization-based methods (Zhu et al., 2021). In comparison, the encoder employed in our work only requires a single forward pass and is never tuned at test time, thus requiring minimal overhead while preserving the benefits of an optimization-based registration method. This is now illustrated in __Appendix C__, where we show fine-tuning performance for VoxelMorph (Balakrishnan et al., 2019) over time.
> We agree that performance gains vary across different sets, which we discuss in the __Discussion (Section 5, paragraphs 2 and 3)__. We discuss the role of the training set size and the effect of noise. In future work, we will train the proposed method on a larger dataset, with the aim to demonstrate the feasibility of zero-shot registration, with test-time optimization reserved for out-of-distribution images. We now state this more explicitly in the __Discussion (Section 5, paragraph 3)__:
>
> _“Future work may therefore include a large set of more diverse training data, or research more expressive encoding architectures. This could further boost zero-shot performance and limit the necessity of test-time optimization to out-of-distribution data.”_
>
> __W3:__ The methodology is not adequately elucidated, with Figures 1 and 2 failing to convey the nuances of the proposed approach.
>
> __A:__ We apologize for any potential unclarity in Figures 1 and 2. We have updated the figure captions to explicitly cover all modules presented in the figures, and hope they explain the specific parts more clearly now. They now read:
>
> __Figure 1:__ Overview of the proposed method. A source image $S$ and target image $T$ are embedded into a combined representation $\overline{e}$ using a trainable convolutional encoder $E$. We subsequently infuse the parameters of two template implicit neural representations (INR) $\phi_{S\rightarrow T \/ T\rightarrow S}$ by modifying their weights using the embedding $\overline{e}$ in the infusion module. This results in two task-informed INRs~($\phi^*$) capable of rapidly fitting the deformable registration of coordinates in $S$ to coordinates in $T$ in a cycle-consistent fashion.
>
> __Figure 2:__ The REINDIR infusion module. For each implicit neural representation (INR) layer $l$, two modulation vectors $\gamma_l$ and $\delta_l$ are inferred from $\overline{e}$. In order to perform pointwise modulation of weight matrix $W_l^\phi$, the $\gamma_l$ vector is broadcasted to $W_l^\gamma$ through repetition ($R$). Meanwhile, bias $\beta_l^\phi$ is directly modulated by the same-size vector $\delta_l$. This allows the model parameters $W_l$ and $\beta_l$ to instantly incorporate motion pattern information between source and target images. This is repeated for all $L$ layers of each INR. Note that no modulation matrices or bias vectors are predicted for the first ($x,y,z\rightarrow a_0$) and final ($a_L\rightarrow dx,dy,dz$) INR layers.

---

### Official Review · Reviewer_pFD7 · 2024-02-29

**Confidence:** 4
**Preliminary Rating:** 3
**Final Rating:** 3.5

**Summary:**

The paper proposes a module that modulates the weight matrices of an registration INR with learned embeddings. This results in a individual initialization of the network, resulting in better zero-shot results, which benefits the inference time.

**Strengths:**

The proposed module improves the initialisation in nearly all cases and outperform the comparison method for two out of three datasets.
The authors show failure cases and hypothesize well, why these cases might have failed.

**Weaknesses:**

While the benefit of the method is shown for the first few optimization steps, the benefit when training until convergence is not evaluated.
The inference time of the embedding network is not taking into account.
It is unclear how dependent the method is on the exact embedding network. An ablation study would be beneficial.

**Detailed Comments:**

Tancik et al. (2021) should be cited.

**Justification Of Final Rating:**

The authors addressed my questions well. The additional evaluation of runtime is helpful to asses the actual benefit of the methods. While the benefit is indeed minor, the author make clear that their method does outperform Reptile when using very few iterations, useful in real-time applications.
My initial concerns still stand. However, the novelty and clear presentation of the method alongside the new runtime analysis justify an **boderline** acceptance.

**Justification Of The Preliminary Rating:**

The method is novel and shows good results. However, the extend of the resulting benefit from the better initialization seems to be minor. I look forward to the response of the authors during the rebuttal to finalise my rating.

**Questions To Address In The Rebuttal:**

Repeating the embedding to gain a properly sized modulation matrix seems arbitrary. This is especially the case since the lengths of the embeddings are fixed. Would it be negative to predict the full modulation matrix?
It seems like the better initialization saves only around ten optimization steps compared to random initialisation, saving about 1 second in inference time, even less when compared with the meta-learning approach. How many steps and/or time is saved through the better initialization when properly training the network until convergence?
How much time does the inference of the embedding network take?

---

> ### Author Response · Authors · 2024-03-16
> **Response to Reviewer pFD7 (1/1)**
>
> We thank the reviewer for their helpful remarks, for which we list our answers on a per-point basis below.
>
> __W1:__ While the benefit of the method is shown for the first few optimization steps, the benefit when training until convergence is not evaluated.
> __Q2:__ It seems like the better initialization saves only around ten optimization steps compared to random initialisation, saving about 1 second in inference time, even less when compared with the meta-learning approach. How many steps and/or time is saved through the better initialization when properly training the network until convergence?
>
> __A:__ To give a sense of the results for longer test-time optimization, we have included an alternate version of Figure 3 in __Appendix C__ that shows results for the NLST set up to 50 optimization steps, rather than 25. This also shows that with Jacobian determinant regularization, the result after optimizing for 2 seconds from Reptile-based initialization is equivalent to the result with the proposed method after 1.34 seconds (i.e. a 33% speed increase).
>
> __W2:__ The inference time of the embedding network is not taking into account.
> __Q3:__ How much time does the inference of the embedding network take?
>
> __A:__ We thank the reviewer for pointing this out; the inference time of the embedding network was indeed not taken into account. This network runs in approximately 62 milliseconds on our hardware. We have added a figure showing the runtimes of the different approaches in __Appendix C__. The results show inference times for the embedding network and test-time optimization steps (with both Jacobian determinant regularization and Bending penalty regularization). We also indicate the number of optimization steps equaling an image embedding forward pass, which is approximately 1.5 steps for optimization with Jacobian determinant regularization and 0.23 steps with Bending penalty regularization.
>
> __W3:__ It is unclear how dependent the method is on the exact embedding network. An ablation study would be beneficial.
>
> __A:__ We understand the reviewer’s comment regarding alternatives for image embedding architectures. We opted to use the encoder part of the Large Kernel U-Net (LKU-Net, Jia et al., 2022), which is listed as one of the best-performing networks on the Learn2Reg NLST leaderboard. The paper shows little differences between convolutional and transformer-based backbones, with their CNN-based LKU-Net performing best. In agreement with this, we expect any sufficiently parameterized model to perform well as an image encoder. Future work may explore the impact of image encoder choice more explicitly. We now include this reasoning in __Appendix A.1__ to clarify our choice of image encoder:
>
> _“The image encoder used in this work is based on the encoder of the LKU-Net architecture. The authors of the paper extensively explore differences between convolutional and transformer-based backbones, and are listed as one of the best-performing algorithms on the Learn2Reg NLST leaderboard. Nevertheless, we expect any sufficiently parameterized model to perform well as an image encoder.”_
>
> __DC1:__ Tancik et al. (2021) should be cited.
>
> __A:__ We now cite Tancik et al. (2021) in the __Introduction (Section 1, paragraph 2)__:
>
> _“At test time, the INR is then able to quickly adapt to new similar tasks (Nichol et al., 2018; Tancik et al., 2021).”_
>
> __Q1:__ Repeating the embedding to gain a properly sized modulation matrix seems arbitrary. This is especially the case since the lengths of the embeddings are fixed. Would it be negative to predict the full modulation matrix?
>
> __A:__ Generating the full INR weight matrices has indeed been proposed in earlier works (Babu et al., 2023), but it would be challenging for the proposed method. Previous INR-based registration methods typically employ weight matrices of 256x256. Predicting such weight matrices from our image encoder bottleneck (with embedding size 1025 in our current work) with an additional linear layer would result in a large number of additional trainable parameters (i.e. over 130 million in that one layer). Given that even our current encoder requires fairly large datasets to generalize properly to new images, we do not expect to have enough data to properly train such a large translation layer, likely resulting in overfitting.

---

> > ### Comment · Reviewer_pFD7 · 2024-03-25
> >
> > The authors addressed my questions well. The additional evaluation of runtime is helpful to asses the actual benefit of the methods. While the benefit is indeed minor, the author make clear that their method does outperform Reptile when using very few iterations, useful in real-time applications.
> > My initial concerns still stand. However, the novelty and clear presentation of the method alongside the new runtime analysis justify an acceptance.

---

### Official Review · Reviewer_YWF6 · 2024-03-01

**Confidence:** 3
**Preliminary Rating:** 3
**Recommendation:** Poster
**Final Rating:** 2.5

**Summary:**

This paper aims to address the cost of running test-time optimization for implicit neural representation (INR) based image registration methods. This paper proposes to use an image encoder to map a pair of test images to an embedding that is used to modulate the INR's weight. The whole framework is trained end-to-end with Reptile [1]. The authors demonstrated that per-example modulation requires a smaller number of test-time optimization steps to achieve the same level of registration performance.


[1] On First-Order Meta-Learning Algorithms

**Strengths:**

- This paper addresses an important problem in image registration using INRs: significant time spent on test-time optimization.
- This paper is overall very easy to understand. The contribution is clear and the experimental results seemed realistic and honest as negative results are also reported.

**Weaknesses:**

- My main concern is that the proposed approach is better than the Reptile baseline when the number of test-time optimization steps is small. The gain is either nonexistent or marginal for larger number of test-time optimization steps, e.g., 10. The registration performance is greatly improved if do a few more optimization steps, implying that realistically people would do more than 10 optimization steps anyways where the proposed approach do not have significant advantage over simple baselines.
- This paper lacks sufficient references on how past works have modulate network weights and how their approach is different. Related, this paper lacks sufficient comparisons to alternative ways to modulate network weights. More details in further comments below.

**Detailed Comments:**

- Abstract: introduce INR before using INR
- Introduction: talk a bit about what are the advantages of INR over optimization / DL based methods.
- Move Section 2 Data to Section 4 Experiments to avoid repeating description of the datasets.
- It would be nice to provide some information of the overhead of using an additional encoder that generates embeddings of image pairs during training and compare that with that of the Reptile baseline.
- Figure 3: It would be helpful to include "Runtime" in additional to "optimization steps' in the x-axis or in caption/text to understand how much time could be saved by using REINDIR vs. Reptile.
- It would be helpful if the authors write a few sentence on the motivation for outputing two modulation vectors, and broadcast one to the size of the 2D matrix through repetition. It would be helpful if the authors provide ablations on why doing so is helpful. For example, compare with (1) just modulate the bias term in each layer, (2) just modulate the 2D matrix in each layer, (3) modulate the matrix term by predicting a 2D matrix instead of a vector then repeat, (4) modulate the weight matrix by addition instead of point-wise product, (5) other modulation methods like [1,2]. The authors should either run the ablations or explain why it's not necessary.
- Related to previous point, it would helpful if the authors explain how prior works modulate neural network weights using hypernetworks differently, e.g., modulate weights using product vs. sum. A few examples of this body of work: [2,3] and I'm sure there are many diferent ways to modulate INR in addition to the 1 reference mentioned in discussion.



[1] FiLM: Visual Reasoning with a General Conditioning Layer

[2] From data to functa: Your data point is a function and you can treat it like one

**Justification Of Final Rating:**

I don't feel the authors sufficiently addressed my concern.

- On the dataset that the proposed method works the best (NLST), it seems that if your test-time budget is 0.1 seconds, then REINDIR yields 30% lower error than the Reptile baseline. However, with a test-time budget >= 0.25 seconds, REINDIR and Reptile essentially yields the same error. Moreover, by just setting a relatively larger test-time budge of 0.5 second, the error of both methods improved significantly (>50%) compared to a test-time budget of 0.25 second. If the motivation of the method is for problems that warrant faster registration, the authors should demonstrate significant performance gain at a certain test-time inference budget that makes sense for a specific/concrete application/task. Currently, the take-away from reading these figure 3 and figure 8 is that the proposed REINDIR is not that different from Reptile.
- The authors are proposing a new hypernetwork that "broadcast the encodings to fill the modulation weight matrices." I don't think it's an unreasonable request to ask for how existing hypernetworks do in the same experimental setup. I agree it's perhaps not enough time to run these additional experiments, and perhaps it's better if the authors could run these experiments and make the current paper more complete.

**Justification Of The Preliminary Rating:**

Despite the paper being easy to read and tried to solve an important problem, the proposed method performs similarly to baseline methods in realistic use cases, e.g., run test-time optimization for >10 steps. Additionally, the paper lacks sufficient reference and necessary ablations on network weight modulation. Therefore, I recommend borderline.

**Questions To Address In The Rebuttal:**

Adress problems raised in Weaknesses and additional comments. Specifically, ablation on modulation proposed and additional reference on how modulation is done in the past.

**Special Issue:**

No

---

> ### Author Response · Authors · 2024-03-16
> **Response to Reviewer YWF6 (1/2)**
>
> We thank the reviewer for their detailed assessment of the paper; we have tried to address all raised concerns, as listed below (note that this response was split in 2 parts due to the character limit).
>
> __W1:__ My main concern is that the proposed approach is better than the Reptile baseline when the number of test-time optimization steps is small. The gain is either nonexistent or marginal for larger number of test-time optimization steps, e.g., 10. The registration performance is greatly improved if do a few more optimization steps, implying that realistically people would do more than 10 optimization steps anyways where the proposed approach do not have significant advantage over simple baselines.
>
> __A:__ Our primary aim is to design a registration method for clinical applications where long optimization times are unfeasible, such as mesh propagation in real-time settings or in registration of long 4D sequences, where large numbers of 3D volumes need to be registered towards each other. In such applications, registration performance within a short temporal window is critical for clinical viability, as is stressed in the __Introduction (Section 1, end of paragraph 2)__.  We agree the performance boost varies per evaluated dataset and is notably correlated with the number of available training examples. In our future work, we will train the proposed method on a larger dataset, with the aim to demonstrate the feasibility of zero-shot registration, with test-time optimization reserved for out-of-distribution images. We now state this more explicitly in the __Discussion (Section 5, paragraph 3)__:
>
> _“Future work may therefore include a large set of more diverse training data, or research more expressive encoding architectures. This could further boost zero-shot performance and limit the necessity of test-time optimization to out-of-distribution data.”_
>
> __W2:__ This paper lacks sufficient references on how past works have modulate network weights and how their approach is different. Related, this paper lacks sufficient comparisons to alternative ways to modulate network weights. More details in further comments below.
>
> __A:__ We agree with the reviewer that there are many ways to modulate weights, and many versions have been proposed over the past years for various applications. However, none have been applied to image registration, which is a peculiar application of INRs: the information density of these representations is much higher than for a typical neural radiance field or implicit shape network (which are generally extremely sparse with respect to their coordinate space). The amount of information required to fully predict a deformation vector field for any given pair of images is highly dependent on the specific subjects being registered: a (near) rigid transformation could be fully represented with an encoding of only six elements, whereas a highly elastic deformation between two small bowel images may contain multiple megabytes of information. This is our reasoning for keeping the amount of information infused in the INRs a free parameter in the method. We plan to investigate the impact of different encoding lengths for different datasets in a follow-up paper, but as this will require extensive additional experiments, we consider this out-of-scope for the current work. We have updated the __Discussion (Section 5, paragraph 5)__ accordingly:
>
> _“Future work may further explore the impact of different encoding lengths and types of embedding infusion.”_
>
> __DC1:__ Abstract: introduce INR before using INR
>
> __A:__ Thank you for pointing this out; the “INR” acronym is now introduced in the first sentence of the Abstract:
>
> _“The use of implicit neural representations (INRs) …”_
>
> __DC2:__ Introduction: talk a bit about what are the advantages of INR over optimization / DL based methods.
>
> A: We have added a short description of the benefits of INRs in the __Introduction (Section 1, paragraph 2)__:
>
> _“This optimization framework has several advantages over both classic optimization-based methods and deep learning-based methods, such as precise regularization of spatial derivatives, robustness to domain shift, and the ability to capture small details in the DVF, …”_
>
> __DC3:__ Move Section 2 Data to Section 4 Experiments to avoid repeating description of the datasets.
>
> __A:__ We thank the Reviewer for this suggestion, this indeed reads somewhat redundantly. We have therefore swapped positions of the Method and Data sections, with the Method Section now being Section 2, and the Data Section being Section 3. The following text has been removed from the start of Section 4, as it is now redundant:
>
> _“We evaluate our method on three chest CT datasets: the DIR-LAB set (10 image pairs) (Castillo et al., 2019), a subset of the NLST set (100 image pairs) (Aberle et al., 2011), and the CBCT set (Balik et al., 2013) (26 image pairs), the latter two as included in the Learn2Reg challenge (Hering et al., 2022).”_

---

> > ### Author Response · Authors · 2024-03-16
> > **Response to Reviewer YWF6 (2/2)**
> >
> > __DC4:__ It would be nice to provide some information of the overhead of using an additional encoder that generates embeddings of image pairs during training and compare that with that of the Reptile baseline.
> >
> > __A:__ Thank you for pointing out this oversight. We have added a figure presenting the runtimes of method variations in __Appendix C__. The results also show inference time for the embedding network and test-time optimization steps (with both Jacobian determinant regularization and Bending penalty regularization). We also indicate the number of INR optimization steps equaling an image embedding forward pass, which is approximately 1.5 steps for optimization with Jacobian determinant regularization and 0.23 steps with Bending penalty regularization.
> >
> > __DC5:__ Figure 3: It would be helpful to include "Runtime" in additional to "optimization steps' in the x-axis or in caption/text to understand how much time could be saved by using REINDIR vs. Reptile.
> >
> > __A:__ See __DC4:__ we now also refer to the runtime benchmark results in the __caption for Figure 3__.
> >
> > Figure 3 caption: _“Registration error over time, comparing random initialization (ccIDIR), learned initialization (Reptile), and the proposed learned initialization with embedding infusion (REINDIR). Median results plotted for 10 different random seeds; solid lines indicate the median, shaded areas indicate the range among seeds. Extended results regarding performance benchmarks and runtimes are listed in Appendix C.”_
> >
> > __DC6:__ It would be helpful if the authors write a few sentence on the motivation for outputing two modulation vectors, and broadcast one to the size of the 2D matrix through repetition. It would be helpful if the authors provide ablations on why doing so is helpful. For example, compare with (1) just modulate the bias term in each layer, (2) just modulate the 2D matrix in each layer, (3) modulate the matrix term by predicting a 2D matrix instead of a vector then repeat, (4) modulate the weight matrix by addition instead of point-wise product, (5) other modulation methods like [1,2]. The authors should either run the ablations or explain why it's not necessary.
> > __DC7:__ Related to previous point, it would helpful if the authors explain how prior works modulate neural network weights using hypernetworks differently, e.g., modulate weights using product vs. sum. A few examples of this body of work: [2,3] and I'm sure there are many diferent ways to modulate INR in addition to the 1 reference mentioned in discussion.
> >
> > __A:__ We originally considered generating the full weight matrices (option 3, as in Babu et al. 2023), but to do so for the same size networks used in previous literature on INR-based registration, this would require a linear layer from the bottleneck size (1025 in our current work) to 2x256x256, which results in over 130 million parameters in that single layer. This is several times larger than our proposed image encoder. Given that even our current encoder requires large datasets to generalize well to new images, we do not have enough data to properly train such a large translation layer, likely resulting in severe overfitting.
> > Though other modulation options (bias only, weight matrix only, activation only, addition, multiplication) are indeed interesting ablations, re-running the experiments is unfortunately infeasible given the limited rebuttal time. We hypothesize that the hypernetworks could indeed already boost performance significantly using shift modulation only, i.e. only modulating the bias vector (either through multiplication or addition), as indicated by Dupont et al. [2]. Though [1] modulates the activations rather than the weight and bias matrices (as also used by Papa et al.), we theorize that having a network modulate both does not negatively impact results - at worst, it could cause minimal additional overhead. We have added a note on this in the __Discussion (Section 5, paragraph 5)__:
> >
> > _“Recent works (Dupont et al., 2022; Papa et al., 2023) proposed to modulate INR activations rather than the weight matrices themselves, which allows the network to condense meaningful sample-specific information into a low number of parameters. Future work may investigate such modulation variations.”_
> >
> > We now also cite FiLM and provide a short description in the __Introduction (Section 1, paragraph 3)__:
> >
> > _“... to directly predict the weights or modulate the activations of an INR. … (Ha et al., 2017; Perez et al., 2018; Babu et al., 2023).”_

---

### Official Review · Reviewer_s7rH · 2024-03-03

**Confidence:** 3
**Preliminary Rating:** 5
**Recommendation:** Best Paper Award
**Final Rating:** 5

**Summary:**

The paper presents REINDIR, a method for enhancing deformable image registration using implicit neural representations (INRs). By infusing image embeddings into INRs, REINDIR optimizes the registration process, significantly reducing optimization time during inference. This approach is demonstrated to be efficient and robust across different datasets, including challenging scenarios with severe domain shifts. The method's innovation lies in its use of meta-learning and embedding infusion, setting a new standard for quick and reliable deformable image registration in medical imaging.

**Strengths:**

The strength of this paper lies in its innovative approach to deformable image registration through the integration of implicit neural representations (INRs) and meta-learning techniques. REINDIR significantly improves the efficiency and accuracy of image registration tasks by reducing optimization time during inference. This is particularly beneficial in medical imaging applications, where quick and precise alignment of images is crucial. The method's robustness across various datasets, including those with severe domain shifts, underscores its potential for broad applicability and its advancement over existing registration methods.

**Weaknesses:**

The paper are well constructed and with no obvious weakness. This paper has sufficient amout of experiments, clear explanation, and the idea is very novel. If possible, the authors can bring part of the experiment details to the main body.

**Detailed Comments:**

N/A

**Justification Of Final Rating:**

After reviewing this paper and other reviewer's suggestion, I think this paper is novel and has solid contributions. If possible, the author can try to polish the work again and provide more comprehensive explanation and experiment in the camera ready version.

**Justification Of The Preliminary Rating:**

This paper has shown a significant amout of work, including the novelty, amout of experiments. The explanation is quite clear, the Figures, Algorithms are clear to read and understand. It is easy to understand this paper within a limited amount of time.

**Questions To Address In The Rebuttal:**

N/A

**Special Issue:**

Yes

---

> ### Author Response · Authors · 2024-03-16
> **Response to Reviewer s7rH (1/1)**
>
> We thank the reviewer for their encouraging assessment of the paper.
>
> __W1:__ The paper are well constructed and with no obvious weakness. This paper has sufficient amout of experiments, clear explanation, and the idea is very novel. If possible, the authors can bring part of the experiment details to the main body.
>
> __A:__ We thank the reviewer for recognizing our work and for their positive feedback. Regarding the experimental details, we agree that it would be useful to address some key aspects of our method more directly in the main body. Therefore, we now include general details about the image encoder (E) and INR modulation in __Section 2.2 paragraph 2 and 3__:
>
> _“The INRs used in this work are SIRENs (Sitzmann et al., 2020) with three hidden layers containing 256 features each. Modulation is performed for the two weight matrices between these hidden layers.”_
>
> _“... first embedded by a convolutional encoder based on the LKU-Net architecture (Jia et al., 2022), which is used to generate modulation vectors.”_
>
> Due to space constraints, remaining extensive experimental details are described in __Appendix A__. Additionally, a more extensive runtime analysis was highly suggested by other reviewers, for which we now show the results in __Appendix C__.

---

### Meta-Review · Area_Chair_Amab · 2024-04-05

**Recommendation:** Accept (Poster)
**Confidence:** 2

**Metareview:**

The authors modulate the weights of a Neural Deformable Image Registration Network via meta-learning. Overall, the reviewers found the paper interesting, but of minimal difference (experimentally) to existing methods. In most situations, the improvement is minimal or non-existent.  Unfortunately, in 2 of the 4 reviews, the reviewer score went down after rebuttal, with the reviewers being unsatisfied. I believe all reviewers here acted in good faith and were thorough.

Overall, the paper paper remains borderline. It may be insightful to communicate at the conference, but I believe it will also be good for the authors to take all this feedback and strengthen the paper.

---

### Decision · Program_Chairs · 2024-04-06

Accept (Poster)